# Attitudes and Behaviors of Physical Activity in Children with Cerebral Palsy: Findings from PLAY Questionnaire

**DOI:** 10.3390/children9070968

**Published:** 2022-06-29

**Authors:** Dai Sugimoto, Amy E. Rabatin, Jodie E. Shea, Becky Parmeter, Benjamin J. Shore, Andrea Stracciolini

**Affiliations:** 1Faculty of Sport Sciences, Waseda University, Tokyo 202-0021, Japan; 2The Micheli Center for Sports Injury Prevention, Waltham, MA 02453, USA; andrea.stracciolini@childrens.harvard.edu; 3Pediatrics Physical Medicine & Rehabilitation, Mayo Clinic, Rochester, MN 55902, USA; rabatin.amy@mayo.edu; 4Department of Orthopedic Surgery, Boston Children’s Hospital, Boston, MA 02115, USA; jodie.shea1@gmail.com (J.E.S.); benjamin.shore@childrens.harvard.edu (B.J.S.); 5Division of Sports Medicine, Boston Children’s Hospital, Boston, MA 02115, USA; becky.parmeter@childrens.harvard.edu; 6Harvard Medical School, Boston, MA 02115, USA; 7Division of Emergency Medicine, Department of Medicine, Boston Children’s Hospital, Boston, MA 02115, USA

**Keywords:** exercise, well-being, CP, pediatrics, disabilities, adaptive sports

## Abstract

To investigate the domains of physical activity in children with cerebral palsy (CP) and to compare these findings to typically developed (TD) children. ***Methods***: A cross-sectional study design. Responses of the four domains in Play Lifestyle and Activity in Youth (PLAY) questionnaire were descriptively analyzed and compared between children with CP (GMFCS I-II) and TD children. ***Results***: Fifty-three children with CP (N = 53, 36 males and 17 females, age of 8.4 ± 1.7 years) and 58 TD children (N = 58, 34 males and 24 females, age of 7.6 ± 1.4 years) participated in this study. In analyses of daily behavior, reported participation in weekly (adaptive) physical education (PE) and sports were more frequent in children with CP (0.6 ± 0.5 days per week) compared to TD children (0.4 ± 0.6 days per week, *p* = 0.040). Outside play time including free play, organized (adaptive) sports and recess were higher in children with CP (2.7 ± 0.8 days per week) than TD children (2.4 ± 0.7 days per week, *p* = 0.022). About motivation/attitudes, a higher proportion of TD children feel sad if they are not able to play sports during the day (74.1%) compared to children with CP (48.7%, *p* < 0.001). ***Conclusion***: Physical activity level was comparable between children with CP and age-matched TD children, while TD children showed higher scores in knowledge and understanding, motivation/attitudes, and physical competence.

## 1. Introduction

The World Health Organization updated the physical activity guidelines in 2020 and recommended, on average, 60 min per day of moderate to vigorous physical activity (MVPA) across the week for children and adolescents [1]. The updated guidelines included various populations such as pregnant and post-partum women, individuals with chronic conditions and disabilities [2]. The inclusion of children with disabilities in the WHO physical activity guidelines is crucial for many reasons. Physical activity for children with disabilities promotes inclusion, minimizes deconditioning, optimizes physical functioning, improves mental health as well as academic achievement, and enhances overall well-being [3]. According to the Centers for Disease Control and Prevention (CDC), children with disabilities are approximately 40% more likely to be obese than typically developing children [4]. Cerebral palsy (CP) is a permanent, non-progressive disorder of the development of movement and posture. In various physical and cognitive disabilities, CP is one of the most commonly diagnosed physical disability of childhood [5,6], and recent prevalence estimates for CP is 3.2 (95%CI: 2.7, 3.7) per 1000 children ages 3–17 years [7].

Due in part to lack of encouragement and limited opportunities to participate in physical activity, exercise, or competitive sports, many children with CP engage in more sedentary and solitary activities, leading to a higher prevalence of overweight and obesity, lower levels of cardiorespiratory fitness, and increasing social isolation [8,9,10]. To counter these negative health consequences, the benefits of physical activity in the CP population was advocated [11,12]. One longitudinal study of CP individuals identified a greater likelihood of being physically active in adulthood when physical activity was performed in mid-teens [11]. Therefore, development of sound physical activity behavior was advocated to your individuals with CP [12].

Investigations related to the physical activity behavior in children with CP are limited. Recent studies that examined exercise behavior in typically developing (TD) children identified four important domains of physical activity which consisted of: daily behavior, knowledge and understanding, motivation/attitudes, and physical competence [13,14]. However, these domains have not been investigated in children with CP. For the aforementioned reasons, it is imperative that we strive to enhance our understanding of physical activity behaviors in children with CP. The purpose of this study was to investigate the four domains surrounding physical activity in children with CP (GMFCS level I-II) and to compare these findings to TD children. It was hypothesized that children with CP would generally show lower scores in the four domains surrounding physical activity compared to TD children.

## 2. Methods

### 2.1. Study Design

A cross-sectional study design was used. The study setting was a pediatric sports medicine clinic at a tertiary-level academic medical center, located in the New England region of the United States. The current study was conducted prior to the COVID-19 pandemic (October 2017 and August 2018). An ethical approval was obtained through Boston Children’s Hospital Institutional Review Board (IRB) prior to commencement of the study (IRB code: P00034682, September 2017). All the participants were informed and consented to participate in this research via assent and consent forms, which were collected before the initiation of this study.

### 2.2. Participants

Participants were either children with CP diagnoses or TD children. Inclusion criteria were (1) age between 6–11 years (≥6 years and ≤11 years) and (2) for children with CP diagnosis, GMFCS I or II classification [15]. Exclusion criteria were (1) age younger than 6 and older than 11 years old and (2) GMFCS level III-V classifications in children with CP. Eligible children with CP were approached by study personnel prior to their appointment at the Cerebral Palsy Clinic at Boston Children’s Hospital. TD children were recruited at a local YMCA near the host study institution.

### 2.3. Procedures

The Play Lifestyle and Activity in Youth (PLAY) questionnaire [13] was used for TD children while a modified PLAY questionnaire was developed and used for children with CP. The PLAY questionnaire [13] was originally designed to capture the four physical activity domains including daily behavior, knowledge and understanding, motivation/attitudes, and physical competence of TD children. The questionnaire included questions that pertain uniquely to each domain as well as some questions that cross over to other domains, recognizing that each component of the questionnaire in isolation will not suffice in painting the full physical activity picture, but rather the synthesis of all the questions together. For example, the compellation of daily behavior questions consisted of screen time, bedtime rules, outdoor play equipment and access.

For this study, the PLAY questionnaire was modified for the investigation of children with CP. For instance, participation in adaptive physical education (PE) class and adaptive sports was included in the modified PLAY questionnaire. Additionally, definitions and examples of the MVPA physical activity options and the organized versus non-organized play/participation were verbally explained to maintain consistency of the key words used in the modified PLAY questionnaire. The PLAY and modified PLAY questionnaires were answered by both parents and caregivers of TD children and children with CP.

### 2.4. Outcome Measures

The primary outcome measures were PLAY questionnaire responses from children with CP and TD children participants. Responses were captured from selected questions from the modified PLAY and PLAY questionnaires in the four domains (daily behavior, knowledge and understanding, motivation/attitudes, and physical competence). The selected questions were chosen because they were pertinent to both children with CP and TD children. Two authors (AR and AS) screened the questions that are applicable for children with CP and TD children.

### 2.5. Statistical Analysis

Descriptive statistics including mean and standard deviations values were used to report physical characteristics of participants. Frequency of the responses in the four domains (daily behavior, knowledge and understanding, motivation/attitudes, and physical competence) was converted to percentages (%) based on the entire proportion. To compare continuous variables between children with CP and TD children, Shapiro–Wilks test was initially performed to examine normality of the data distribution. When the Shapiro–Wilks test showed a normal distribution of the data, an independent sample *t*-test was used to compare continuous variables. When a non-normal distribution was found by the Shapiro–Wilk test, Mann–Whitney U test was utilized. For both independent sample *t*-test and Mann–Whitney U test, two-tailed tests were chosen. For categorical variables, a two-sided Fisher’s exact test was performed. Statistical significance of *p* < 0.05 was used for Shapiro–Wilks tests and all comparisons.

## 3. Results

### 3.1. Demographics

There were 53 children with CP (36 males and 17 females) and 58 TD children (34 males and 24 females). Physical characteristics, grades in school, and race/ethnicity are presented in Table 1. The fifty-three children with CP consisted of level I [N = 26 (49.1%)] and level II [N = 26 (49.1%)] of the GMFCS level, and there was one child with CP who did not wish to share this information [N = 1, (0.8%)]. Furthermore, clinical diagnoses of the children with CP were expressed in Figure 1.

### 3.2. Daily Behavior

Although there were no differences in the average number of physically active days during weekdays and weekend between children with CP and TD children (Table 2), the reported number of physically active days per week in the past week was greater in children with CP (4.7 ± 2.0 days per week) than TD children (3.2 ± 2.2 days per week, *p* < 0.001, Table 2). In addition, reported participation in weekly physical education class was more frequent in children with CP (0.6 ± 0.5 days per week) compared to TD children (0.4 ± 0.6 days per week, *p* = 0.040, Table 2). Moreover, outside play time including free play, organized (adaptive) sports, and recess was higher in children with CP (2.7 ± 0.8 days per week) than TD children (2.4 ± 0.7 days per week, *p* = 0.022, Table 2). Hours of sleep reported during the weekday were shorter in children with CP (8.7 ± 1.0 h per day) than TD children (9.3 ± 1.1 h per day, *p* = 0.014, Table 2). Moreover, hours of TV watched per weekend were shorter in children with CP (1.7 ± 1.1 h per day) compared to TD children (2.6 ± 1.3 h per day, *p* < 0.001, Table 2).

### 3.3. Knowledge and Understanding

In response to the question, “Do you think you are familiar with general health tips and related information?”, a lesser proportion of children with CP and caregivers responded “yes” (38.9%) compared to TD children and parents (62.1%, *p* = 0.034, Table 3). When study participants were asked the question “What is the longest amount of time per day children should look at a screen including cell phone, computer game, and TV?”, the screen time duration responded by children with CP and caregivers was longer (1.7 ± 1.2 h per day) than TD children and parents (0.6 ± 0.8, *p* < 0.001, Table 3).

### 3.4. Motivation/Attitudes

In response to the question, “Do you feel sad if you are not able to play (adaptive) sports or run around during the day?”, a fewer proportion of children with CP gave feedback of “yes” (48.7%) compared to TD children (74.1%, *p* < 0.001, Table 4).

### 3.5. Physical Competence

When study participants were asked the question, “Do you feel physically competent compared to your peers?”, a lesser proportion of children with CP answered “yes” (49.1%) compared to TD children (94.8%, *p* < 0.001, Table 5). Additionally, in response to the question, “Has your teacher ever been concerned about your handwriting?”, a greater proportion of children with CP and caregivers replied “yes” (63.2%) than TD children and parents (24.1%, *p* < 0.001, Table 5).

## 4. Discussion

The purpose of this study was to investigate physical activity behavior in children with CP in relation to TD children. The hypothesis was that children with CP would generally demonstrate lower scores in the four domains surrounding physical activity compared to TD children. According to our results, the daily behavior (Table 2) was comparable between children with CP and TD children, while TD children showed greater scores in the rest of the three domains [knowledge and understanding (Table 3), motivation/attitudes (Table 4), and physical competence (Table 5)] than children with CP. Thus, the hypothesis was mostly supported. One salient finding from this study came from a domain of daily behavior. The level of physical activity reported was comparable between children with CP and TD children except for physical activity days in the past week (Table 2). However, children with CP participated in adaptive PE, sports, and recess more frequently than TD children (Table 2). These differences were subtle (Table 2), but it is worth noting that children with CP reported slightly more time in their PE and outside play compared to TD children. These findings were different to previous reports regarding physical activity level in young CP population in conjunction with TD children [16,17,18]. Comparing teenagers with CP (GMFCS level I–III) and their age-matched TD peers, a study from South Africa reported step counts were comparable [17]. However, a report from the Netherlands that compared physical activity level between children with CP (age 5–7 years, GMFCS level I-IV) and the same age TD peers showed lower physical activity level in children with CP than their counterpart TD children [18]. Finally, one systematic review study evaluated young people with CP and concluded that their habitual physical activity was 13–53% lower than their TD peers [16]. However, in this systematic review, one of the six studies that had the most CP participants (N = 112) included all GMFCS levels (I–V), which might have influenced the lower physical activity outcome because an association between greater physical impairment and reduced physical activity level was reported in children with CP [19].

One novel finding associated with physical activity was observed in motivation/attitudes (Table 4). Although both children with CP and TD children dominated similar percentages of enjoyment and happiness in (adaptive) PE and sports, there was a substantial difference in sadness (Table 4). Approximately, three out of four (74.1%) of TD children showed sadness if they cannot play sport during the day (Table 4). However, the rate was about one in two (48.7%) in children with CP (Table 4). Previous research has identified a connection between happiness and physical activity in children and adolescents with CP [20], where physical activity is a predictor for happiness as well as physical and social well-being in adolescents with CP [20]. Based on this study report [20], it is logical to assume physical activity is important for children with CP in their daily livings, which is shown in motivation/attitudes data (Table 4). However, specific reasons pertaining to the response to this question in the children with CP are unclear. This may be an area of future study.

Another interesting finding was a difference in knowledge related to screen time (Table 3). Children with CP and parents of children with CP responded 1.7 h per day of screen time (including cell phone, computer games, and TV) as the longest amount of recommended screen time, while the feedback of the same question from TD children and parents of TD children was 0.6 h per day (Table 3). It is plausible to surmise that children with CP habitually use screens more than TD children in their daily lives. However, according to our daily behavior comparisons, cell phone use and computer games were generally comparable between the two groups or slightly higher in TD children (Table 2). About TV times, TD children actually showed longer time than children with CP, especially during weekend (Table 2). This disparity indicates that the knowledge presented in this study is not necessarily reflective to their daily behavior, or there may be unknown reasons for the gap. While video games and screen time may be viewed as a sedentary activity, one study suggested that video games may provide a light-moderate physical activity opportunity to children with CP [21]. Additionally, one randomized controlled trial reported that video game training enhanced balance ability better than traditional balance training in children with CP (GMFCS levels I–III) [22]. Understanding the benefits of screen time in children with CP is a new frontier of research.

## 5. Limitation

There were a couple shortcomings in this study. First, the current study data were collected from one geographic region. It is known that access to recreational, leisure, and sport facilities is a crucial parameter for physical activity [23], which might have impacted the results of the current study. Second, nearly half of the TD children and parents of TD children did not answer the race/ethnicity question. Parents of TD children came to the YMCA to drop off or pick up their children, and in the limited timeframe, this question was often neglected. Third, no study was performed to validate the questionnaires used in this study. Both PLAY and modified PLAY questionnaires were developed by scientists, physicians, and therapists in a field of physical activity and pediatric medicine; thus, all questions are clinically relevant. Validation of the questionnaires would have further substantiated the current data. Fourth, there were only two comparisons in physical competence. It was challenging to find appropriate comparisons between the two groups except for the two questions (competence related to peers and quality of handwriting, Table 5). Finally, the current study is a cross-sectional in nature. Long-term effect of the physical activity in children with CP is unknown.

## 6. Conclusions

Notable differences in reported attitudes and behaviors surrounding physical activity were found between children with CP and TD children. In short, the current study identified that the physical activity level of children with CP was comparable with age-matched TD children. Participation of (adaptive) PE and sports was greater in children with CP than TD children. A fewer proportion of children with CP responded “sad” if there was no opportunity of playing sports during the day compared to the TD children. Screen time of the children with CP was about the same duration as TD children. Future studies are warranted to evaluate a long-term effect of physical activity in children with CP.

## Figures and Tables

**Figure 1 children-09-00968-f001:**
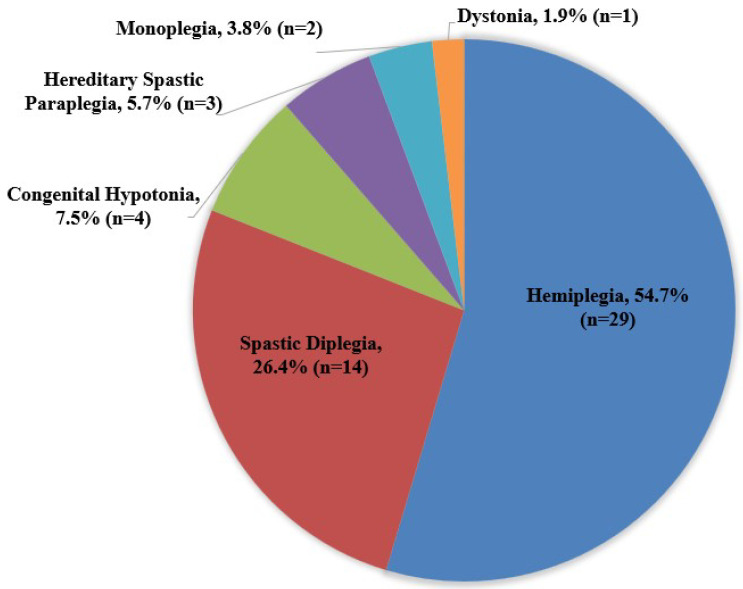
Diagnoses of Children with CP.

**Table 1 children-09-00968-t001:** Demographics of children with CP and TD children.

Variables	Children with CP N = 53	TD Children N = 58
Physical Characteristics		
Age (years)	8.4 ± 1.7	7.6 ± 1.4
Height (cm)	129.5 ± 13.5	126.5 ± 13.4
Weight (Kg)	29.6 ± 9.1	29.4 ± 10.7
BMI	17.3 ± 3.1	18.3 ± 5.4
Sexes		
Male	36 (67.9%)	34 (58.6%)
Female	17 (32.1%)	24 (41.4%)
Grades in School		
0	7 (13.2%)	5 (8.6%)
1	11 (20.8%)	15 (25.9%)
2	5 (9.4%)	15 (25.9%)
3	10 (18.9%)	12 (20.7%)
4	7 (13.2%)	4 (6.9%)
5	6 (11.3%)	6 (10.3%)
6	6 (11.3%)	0 (0.0%)
7	1 (1.9%)	1 (1.7%)
Race/Ethnicity		
African American	4 (7.6%)	3 (5.2%)
Asian	4 (7.6%)	2 (3.5%)
Caucasian	41 (77.4%)	22 (37.9%)
Latino	0 (0.0%)	0 (0.0%)
Native American	1 (1.9%)	0 (0.0%)
Others	2 (3.8%)	4 (6.9%)
Prefer not to respond	1 (1.9%)	0 (0.0%)
No response	0 (0.0%)	27 (46.6%)

**Table 2 children-09-00968-t002:** Comparisons of Daily Behavior between children with CP and TD children.

Questions	Children with CP	TD Children	*p*-Values
**Level of Physical Activity**			
On average, how many days were you physically active during weekdays? (Days)	2.5 ± 1.5	2.6 ± 1.8	0.937
On average, how many days were you physically active during weekend? (Days)	3.2 ± 1.9	3.0 ± 1.6	0.567
How many days were you physically active in the past week? (Days)	4.7 ± 2.0	3.2 ± 2.2	0.001 *
**PE and Outside Play Participation**			
On average, how many days a week do youparticipate in (adaptive) PE or gym class? (Days)	0.6 ± 0.5	0.4 ± 0.6	0.040 *
On average, how many hours per day do you play outside including free play, organized (adaptive) sports and recess? (Hours)	2.7 ± 0.8	2.4 ± 0.7	0.022 *
**Cell Phone Time**			
On average, how many hours per day do you use cell phone during weekdays? (Hours)	0.5 ± 0.8	0.6 ± 0.8	0.563
On average, how many hours per day do you use cell phone during weekend? (Hours)	0.7 ± 1.2	1.0 ± 1.4	0.224
**Computer Game Time**			
On average, how many hours per day do you play computer game during weekdays?(Hours)	0.4 ± 0.7	0.6 ± 1.1	0.214
On average, how many hours per day do you play computer game during weekend? (Hours)	0.5 ± 0.8	0.8 ± 1.2	0.077
**TV Time**			
On average, how many hours per day do you watch TV during weekdays? (Hours)	1.3 ± 0.7	1.6 ± 1.2	0.146
On average, how many hours per day do you watch TV during weekend? (Hours)	1.7 ± 1.1	2.6 ± 1.3	0.001 *
**Sleep Time**			
On average, how many hours per day do you sleep during weekdays? (Hours)	8.7 ± 1.0	9.3 ± 1.1	0.014 *
On average, how many hours per day do you sleep during weekend? (Hours)	8.9 ± 1.1	9.3 ± 1.1	0.081

Values are mean ± standard deviation. * *p* < 0.05 by independent sample *t*-tests (two-tailed).

**Table 3 children-09-00968-t003:** Comparisons of Knowledge and Understanding between children with CP and TD children.

Questions	Children with CP	TD Children	*p*-Values
**Knowledge**			
Do you think you are familiar with general health tips and related information? (Yes response)	38.9%	62.1%	0.034 *
Do you ask your parents for health tips, health information, or physical activity questions? (Yes response)	44.9%	48.3%	0.848
What is the longest amount of time per day children should look at a screen including cellphone, computer game, and TV? (Hours)	1.7 ± 1.2	0.6 ± 0.8	0.001 *
**Understanding**			
To improve fitness/get stronger, what would you be the best thing to do? (“exercise more” response) ^α^	68.2%	81.0%	0.164
If you enhance aerobic/cardiovascular/endurance fitness, what things you can do more? (“can pump blood more” response) ^β^	46.8%	50.0%	0.306

Values are percentages (%) of each group and mean ± standard deviation. * *p* < 0.05 by independent sample *t*-tests (two-tailed) and Fisher’s exact tests (two-sided) α: Other given choices were: (a) read a book, (b) wait until you are older, and (c) watch a video lesson including coach teaching. β: Other given choices were (a) how well the muscle can push, pull or stretch, (b) having a healthy weight for our height, and (c) our ability to do sports that we like.

**Table 4 children-09-00968-t004:** Comparisons of Motivation/Attitudes between children with CP and TD children.

Questions	Children with CP	TD Children	*p*-Values
**Motivation/Attitudes**			
Do you enjoy participating in (adaptive) PE, gym class, and sports? (Yes response)	94.0%	96.6%	0.659
Do you feel happy when you are able to play (adaptive) sports or run around during the day? (Yes response)	97.2%	98.3%	0.782
Do you feel sad if you are not able to play (adaptive) sports or run around during the day? (Yes response)	48.7%	74.1%	0.001 *

Values are percentages (%) of each group. * *p* < 0.05 by Fisher’s exact tests (two-sided).

**Table 5 children-09-00968-t005:** Comparisons of Physical Competence between children with CP and TD children.

Questions	Children with CP	TD Children	*p*-Values
**Physical Competence**			
Do you feel physically competent compared to your peers? (Yes response)	49.1%	94.8%	0.001 *
Have your teacher ever been concerned about your handwriting? (Yes response)	63.2%	24.1%	0.001 *

Values are percentages (%) of each group. * *p* < 0.05 by Fisher’s exact tests (two-sided).

## Data Availability

Most of the relevant data were presented in the manuscript. The data are not publicly avaiable due to privacy of all participants.

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
