# Peer review of "Attitudes and Behaviors of Physical Activity in Children with Cerebral Palsy: Findings from PLAY Questionnaire"

_children, 2022, doi:10.3390/children9070968_

Round 1

Reviewer 1 Report

Dear Authors

The work presents a study of a selected range of cases, where the impact of physical activity in children after cerebral palsy and healthy children is analyzed, and an attempt is made to compare them. It is a continuation of the design, but also an extension of research conducted by other authors in this field. It is also the use of the research methodology previously proposed by one of the authors. Perhaps the following comments will contribute gain to the work great value.

1. In the abstract, the authors use abbreviations. One of them, i.e. PE is not explained like the previous ones. Here, in Conclusion, there is a generalised, unauthorised statement. It should be specified, to which population this applies.

In this kind of researches the better is using conclude in the following form: there is no basis at the confidence level of p to reject the hypothesis ….

2. A more detailed description of the statistical analysis would be advisable, taking into account:

a. Formulation of hypotheses. Further, in the work the authors use the term "differences". After the hypotheses are clearly formulated, it will be clear what this term ("differences") means.

b. Justifying the choice of test types and their precise naming (whether the test for two dependent or independent samples, two-tailed or one-tailed, etc.)

c. Sequence of tests (the t-test is replaced before the Mann-Whitney U test).

3. From the above remarks and the lack of formulation of the hypothesis, there are some ambiguities in the analysis of the obtained results. The authors simply quote the results shown in the tables. They use the statement that a certain value in CP children is greater than in TD children, or in another case, other values vice versa. This can be seen in the tables. It is the p-value that matters, and the hypothesis under test. All this was certainly what the authors have done but it is not clearly presented in the body of the paper.

Editorial remarks: 

  1. The important information is that a "blind review", the rest ie XXXXXXXXXXXX is irrelevant. Can be removed from the text.
  2. Normally, in parentheses a literature / bibliography citation  after the dot ending the task, is used (if applicable to this sentence). Otherwise, it suggests the the citation relation with the next sentence.
  3. Correcting the editing of the text. It is suggested to put the cite after the period and in one parentheses) line 52, page 1.

Author Response

General Comments: The work presents a study of a selected range of cases, where the impact of physical activity in children after cerebral palsy and healthy children is analyzed, and an attempt is made to compare them. It is a continuation of the design, but also an extension of research conducted by other authors in this field. It is also the use of the research methodology previously proposed by one of the authors. Perhaps the following comments will contribute gain to the work great value.

Authors’ Response:  We appreciate your valuable input, which significantly enhanced a quality of this manuscript compared to the previous version.  The changes we made were highlighted yellow in the revised manuscript.

Comment 1. In the abstract, the authors use abbreviations. One of them, i.e. PE is not explained like the previous ones. Here, in Conclusion, there is a generalised, unauthorised statement. It should be specified, to which population this applies.

In this kind of researches the better is using conclude in the following form: there is no basis at the confidence level of p to reject the hypothesis

Response 1: Thank you for your suggestions. We spelled out “physical education” before we abbreviated it as PE in the abstract.

Also, based on your input, we reworded our conclusion in the abstract with emphasis of the population applied (age of participants in this study), the hypothesis phrased, and the outcome delivered by analyses. 

Comment 2. A more detailed description of the statistical analysis would be advisable, taking into account:

Comment a. Formulation of hypotheses. Further, in the work the authors use the term "differences". After the hypotheses are clearly formulated, it will be clear what this term ("differences") means.

Response a: Thank you for your input.  We added a hypothesis after our purpose statement. Also, as you suggested, we realized how important to make the “difference” clear.  So, our hypothesis is directional, which further helped developing quality discussion.

Comment b. Justifying the choice of test types and their precise naming (whether the test for two dependent or independent samples, two-tailed or one-tailed, etc.)

Response b: Thank you for your recommendation. Based on your recommendation, we stated specific statistical test used in the analysis process. Also, we added the statistical tests used in the analysis underneath of each table.

Comment c. Sequence of tests (the t-test is replaced before the Mann-Whitney U test).

Response c: Thank you for your input. Based on your input, we revised statistical analysis section with sequence of every statistical tests we took.

Comment 3. From the above remarks and the lack of formulation of the hypothesis, there are some ambiguities in the analysis of the obtained results. The authors simply quote the results shown in the tables. They use the statement that a certain value in CP children is greater than in TD children, or in another case, other values vice versa. This can be seen in the tables. It is the p-value that matters, and the hypothesis under test. All this was certainly what the authors have done but it is not clearly presented in the body of the paper.

Response 3: Thank you for your valuable feedback. We added a directional hypothesis and revised statistical analysis section based on your input, which also helped developing a quality discussion and concluding our findings more scientifically rigorous manner.   

Editorial Remarks:

Comment 1. The important information is that a "blind review", the rest ie XXXXXXXXXXX is irrelevant. Can be removed from the text.

Response 1: Thank you for your input. We removed XXXXXXXXX from this manuscript.

Comment 2. Normally, in parentheses a literature / bibliography citation  after the dot ending the task, is used (if applicable to this sentence). Otherwise, it suggests the the citation relation with the next sentence.

Response 2: Thank you for your input. We made the suggested changes for every citation.

Comment 3. Correcting the editing of the text. It is suggested to put the cite after the period and in one parentheses) line 52, page 1.

.
Response 3: Thank you for your comment. It was indeed awkward. We edited citations on the line 52, page 1. 

Reviewer 2 Report

Manuscript reflects information about physical activity behaviors in cerebral palsy children and typically developed children.

-        Study design section has ethical considerations information, this section will be written in other section after statistical analysis.

-        Which differences are between PLAY questionnaires and PLAY modified questionnaire? Both are validated?

-        Name of the figure must be written after the figure.

-        Daily behavior consists in standard questions? Or was it specifically for this study? If it is the PLAY questionnaire, please mention that.

Methodology must be improved to a better understanding of the results. Discussion is quite good, but more references should be added to support your results

Author Response

General Comments: Manuscript reflects information about physical activity behaviors in cerebral palsy children and typically developed children.

Authors’ Response: Thank you for your comments.  The changes we made based on your input were highlighted yellow in the revised manuscript.

Comment 1. Study design section has ethical considerations information, this section will be written in other section after statistical analysis.

Response 1: Thank you for your input. All authors discussed where we should move the ethical consideration information (IRB approval) based on your input.  Since you suggested, we tried moving the part after statistical analysis section.  However, we felt it would divert the flow from statistical analysis to result section. After a few email exchanges, we believe that the ethical consideration may be best under study design.

However, we, of course, respect your input.  Thus, we changed the orders of the sentences within the section. We hope revised version enhanced clarity in the section.

Comment 2. Which differences are between PLAY questionnaires and PLAY modified questionnaire? Both are validated?

Response 2: Thank you for your questions. PLAY questionnaire was used for TD children while PLAY modified questionnaire was made for children with CP. In order to enhance clarity, we revised the method section. 

Both PLAY questionnaire and PLAY modified questionnaire were not validated. This questionnaire was originally developed by scientists and physicians (one of them is a senior author of this paper) who have extensively studied physical activity level in TD children.

Then, the PLAY questionnaire was modified by a physician (2nd author of this study) who dedicated her clinical time to children with CP.    

We feel that all questions in both PLAY and modified PLAY questionnaires are clinically important. Also, a study used the PLAY questionnaire was recently published (PMID: 35212171). However, validate study was not performed. Thus, we added this point in our limitation section.

Comment 3. Name of the figure must be written after the figure.

Response 3: Thank you for your recommendation. We moved the name of the figure at the bottom instead of the top.

Comment 4. Daily behavior consists in standard questions? Or was it specifically for this study? If it is the PLAY questionnaire, please mention that.

Response 4: Thank you for your questions. Physical activity related questions such as “On average, how many days were you physically active during weekdays?” are standard. However, we aimed to capture “daily behaviour” beyond just physical activity level. Some other factors may be related to physical activity level in daily basis. For instance, there were several questions related to outdoor play equipment and access in PLAY questionnaire. We believed that outdoor equipment and access may be impactful for physical activity level.

Now, the outdoor play equipment and access factors in children with CP were not as influential as TD children; hence, we removed those questions in modified PLAY questionnaire and comparisons.

To enhance clarity, we added several sentences in procedure section in method.

Comment 5. Methodology must be improved to a better understanding of the results. Discussion is quite good, but more references should be added to support your results

Response 5: Thank you for your input. We made a few edits in method.  Also, we added a few sentences in discussion.

Round 2

Reviewer 1 Report

Dear Authors

In new version of the manuscript the reviewer's suggestions are considered.

Sincerely Yours